# New Local Drug Delivery with Antibiotic in the Nonsurgical Treatment of Periodontitis—Pilot Study

**Aleksandra Sender-Janeczek, Jacek Zborowski \*, Małgorzata Szulc** 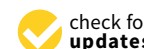 **and Tomasz Konopka**

Department and Division of Periodontology Wroclaw Medical University ul. Krakowska 26, 50-425 Wrocław, Poland; aleksandra.sender-janeczek@umed.wroc.pl (A.S.-J.); malgorzata.szulc@umed.wroc.pl (M.S.); tomasz.konopka@umed.wroc.pl (T.K.)

\* Correspondence: jacek.zborowski@umed.wroc.pl

**Abstract:** Combination of the classical subgingival instrumentation (scaling and root planing procedure, SRP) with an antibiotic administered to periodontal pockets in a suitable medium is a promising alternative protocol of nonsurgical periodontal treatment. It enables obtaining the long-term minimum drug concentration inhibiting the development of periopathogens. Objectives: Clinical and microbiological evaluation of periodontal pockets two months after single application of a gel containing piperacillin and tazobactam (Gelcide)®in relation to the nonsurgical treatment procedure (SRP). Materials and methods: Ten patients aged 24–56 years (mean 39.5) with chronic periodontitis, nonsmokers with acceptable oral hygiene and no classical exclusion criteria were qualified for treatment. In the maxilla area, SRP was performed and the assessed gel was inserted to two randomly selected adjacent periodontal pockets. Clinical evaluation included the assessment of bleeding on probing (BoP), pocket depth (PD), and clinical attachment loss (CAL) at six measurement points. A microbiological examination with the use of PET deluxe diagnostic kit in the drug-administered pockets and symmetrically in two pockets on the other side of the dental arch was performed. The examination was conducted before the treatment and two months later. Results: Two months after the treatment, a significant improvement in all analyzed clinical parameters was observed. However, the extent of this improvement did not differ significantly between the compared treatment methods. No statistically significant differences were found in the number of bacteria before and after the treatment, except for a significant decrease in the number of Micromonas micros (2957 vs. 589, p = 0.028) and a higher number of the green complex bacteria Capnocytophaga gingivalis (5439 vs. 2050, p = 0.041) after antibiotic had been used. Conclusion: No significant clinical and microbiological differences were found after additional administration of gel with piperacillin and tazobactam in relation to SRP in the preliminary study.

**Keywords:** periodontitis; SRP; LDD; gelcide

## 1. Introduction

In the light of the latest classification adopted during the World Panel for Periodontal Classification in 2017, periodontitis is a chronic, multifactorial, inflammatory disease associated with dysbiotic biofilm of the plaque, and is characterized by progressive destruction of dental attachment apparatus [1]. The most characteristic features of the disease include clinical attachment loss (CAL), pocket deepening (PD), bleeding on probing, and loss of bone tissue which can be radiologically observed [2].

Periodontitis constitutes a significant social problem due to the incidence of the disease, and the consequences associated with failure to take up treatment. Loss of teeth that takes place over the course of periodontitis results in the loss of prosthetic support areas, and significantly affects the quality of life of patients as well as the way they function in the society.

Epidemiological studies from 2013 show that 26% of Warsaw residents aged 35–44 require specialist periodontal treatment [3]. In the nationwide studies, the ratio was 16% in the same age group [4] and 19.7% in the group of people aged 65–74. In a 2012 study conducted by Konopka et al. [5], according to the definition of Page and Eke, periodontitis was diagnosed in 23.1% of the patients; in 1.6% of them the disease was severe. According to the definition by Offenbacher et al., 41.2% of the examined patients were diagnosed with periodontitis (0.7% of p1, 22.7% of p2, and 17.8% of p3). In the studies conducted in 2017 [6], the incidence of periodontitis according to DCD/AAP among the residents of Wrocław and Oława aged 65–74 was 30.7% in the cpi4 group, i.e., 47.9, out of which the disease was severe in 21.9 cases [6].

The scaling and root planing procedure (SRP) [7] is considered to be the gold standard in nonsurgical treatment of periodontitis. However, subgingival instrumentation does not eliminate all pathogens from the pocket, therefore reinfections and relapses are very common, especially in the case of deep pockets. One of the promising alternative procedural protocols is a combination of classical subgingival instrumentation (SRP) and intrapocket antibiotic administration in a suitable carrier, which makes it possible to obtain long-term minimum concentration of the drug inhibiting periopathogen development.

Such procedure is known as local drug delivery (LDD). There are many limitations to general antibiotic therapy on periodontal grounds, and it is accompanied by many adverse effects. For this reason, in 1979 Goodson postulated local application of tetracycline [8] administrated into the pockets in the form of vinyl acetate copolymer fibers (Actisite). Local application reduces or eliminates some adverse effects associated with general antibiotic therapy, especially bacterial resistance and allergic reactions. Moreover, concentrations of the active substance in LDD in the area of administration considerably exceed minimal inhibitory concentration (MIC) and enable administration to the particular area that requires treatment [9]. Since 1979, many preparations that release antibiotic active substances and can be applied into the pockets have been introduced to the market and to clinical use. They differ in terms of the carrier, the active substance and the form of administration. Currently, preparations with tetracycline (2 mg), minocycline in 2% concentration and doxycycline in 10% concentration have been in use. These preparations consist of one or two components and can be easily administered to the periodontal pockets.

One of the new forms of LDD for intrapocket application that has been recently introduced is Gelcide. It is a patented mixture that consists of piperacillin and tazobactam combined by a carrier. The preparation is packaged into two vials. After mixing the contents of the vials, the preparation gelifies, and forms a coating that seals the periodontal pocket. Thanks to these properties, the active substances of the product should be retained in the area to which the preparation was administered. The components of Gelcide are slowly released in concentrations exceeding MIC over the period of 8 to 10 days. The coating is permeable, but it is insoluble to fluids—it protects the pocket from exposure to bacteria of the oral cavity and from further irritation and infection.

## 2. Aim

The aim of the paper was to assess the effectiveness of the activity of piperacillin and tazobactam combination in the form of an intrapocket-administered chemotherapy. Its influence on the course of treatment, microbiome of the pocket, healing rate, adverse effects or lack of thereof, is in comparison with the gold standard of periodontitis treatment—the SRP procedure.

## 3. Materials and Methods

Ten patients aged 24 to 56 (the average age was 39.5, 5 males, 5 females) with general periodontitis (loss of CAL on interproximal surfaces, CAL in >30% of teeth, with moderately symmetric bone loss in pantomographic image) were qualified for the study. The patients were healthy nonsmokers who had not used antibiotics within the period of 6 months prior to the study, were not allergic to penicillin, and had at least 10 teeth in the jaw. The excluded criteria were also diabetes mellitus and hypertension.

The second stage of periodontitis was diagnosed in 1 case, the third stage in 6 cases, and the fourth in 3 cases. Degree distribution was 7 from B and 3 from C. All these individuals provided written consent for the proposed treatment protocol. The study was granted approval KB-600/2019 by the Bioethical Committee of Wrocław Medical University.

The procedure would begin with providing instructions on oral hygiene (the obtained average API value was below 20%). Subsequently, machine SRP (ultrasonic scaling with subgingival tips) under nerve block anesthesia in the mandible was performed during two visits. Examination of clinical parameters was conducted by one standardized periodontist (TK). The examinations determined parameters such as bleeding on probing (BoP), clinical attachment location (CAL), and periodontal depth (PD) in six measurement points around each tooth, except for the third molars. Then, microbiological material was collected from two symmetric pockets with similar PD, located on both sides of the dental arch; sterile filter paper was placed in the pockets for 30 s, and the samples were then transferred to transport test-tubes. During the same visit, the periodontist (ASJ) conducted analogous SRP under anesthesia in the jaw. At the end of the visit, a third periodontist (JZ) introduced Gelcide into the periodontal pockets, in accordance with the instructions of the manufacturer, applying randomization to odd-numbered patients in the order of their applications on the left, and even-numbered patients on the right. The randomization was known to only one person (JZ). The patients were informed about the possibility of attending an additional follow-up visit, if any adverse effects associated with the conducted treatment were noticed. If no such effects occurred, the follow-up visits took place one, four, and eight weeks after the initial examination. Clinical examination was conducted again at the last follow-up visit, and the material for microbiological tests (TK) was collected from the same pockets.

During microbiological testing, the PET deluxe test was applied. The test uses the PCR method in real time that makes it possible to quantify the total bacterial count (TBC) in the pocket as well as the red-complex bacteria (*Porphyromonas gingivalis*—P.g., *Treponema denticola*—T.d., and oraz *Tannerella forsythia*—T.f.), *Aggregatibacter actinomycetemcomitans*—A.a.), orange-complex bacteria (*Fusobacterium nucleatum*—F.n., *Prevotella intermedia*—P.i., and *Micromonas micros*—P.m., *Eubacterium nodatum-E.n.*), and green-complex bacteria (*Capnocytophaga gingivalis*—C.g.).

Due to lack of normality of distribution of the analyzed variables, nonparametric tests were used in statistical analysis: the Mann–Whitney U test for nonrelated variables and the Wilcoxon T-test for related variables and for groups below 25 participants. $p \leq 0.05$ was adopted as the significance level. The Statistica package, version 13.1, was used.

## 4. Results

In the vast majority of the examined patients, no symptoms of adverse effects related to the treatment were observed during the scheduled follow-up visits. The protective layer which, according to the manufacturer, was supposed to protect the patients from rapid decrease in drug concentration in the pocket within 8–10 days was unnoticeable. At the last follow-up visit, purulent effusion in the area of drug application was found in one of the patients; inflammatory erythema and exfoliative gingivitis were found in another patient, also in the area of drug application.

Table 1 presents a summary of clinical examination results for the jaw and, separately, for the pockets from which microbiological material was collected and into which the assessed gel was administered, as well as the pockets which were treated using solely SRP (the control group). In the jaw, the treatment resulted in significant decrease in the BoP index (−33%), shallowing of the average PD (−0.45 mm) and PD on the interproximal surfaces (−0.55 mm), as well as decrease in the number of areas with CAL loss ≥5 mm (improvement of the median by 11.5). In the pockets treated using the compared methods, the researchers found similar, considerable reduction in BoP (−49.2% vs. 47.5% for SRP) as well as shallowing of the average PD (0.82 vs. 0.83 mm for SRP) and, additionally, considerable improvement of attachment location in the control pockets (−0.81 mm). Two months after the treatment, no differences between the pockets treated with the use of the evaluated methods were found in terms of the assessed clinical parameters.

**Table 1.** Results of the clinical studies.

| Place of the Test | BoP | PD śr | PD | Mediana PD > 5 mm | Mediana CAL1-2 | Mediana CAL3-4 | Mediana CAL ≥ 5 |
|---|---|---|---|---|---|---|---|
| Upper Jaw | 63.3 ± 22.3 | 4.23 ± 0.92 | 4.86 ± 1.15 | 10 | 7.5 | 13 | 26 |
| | 30.3 ± 12.1 | 3.78 ± 0.97 | 4.31 ± 1.12 | 5.5 | 11 | 7.5 | 14.5 |
| | *p* < 0.000 | *p* = 0.049 | *p* = 0.047 | *p* = 0.59 | *p* = 0.17 | *p* = 0.44 | *p* = 0.012 |
| Pockets SRP+A | 87.5 ± 20.9 | 5.0 ± 1.04 | CAL śr 3.96 ± 1.36 | | | | |
| | 38.3 ± 12.5 | 4.18 ± 0.92 | 3.1 ± 2.02 | | | | |
| | *p* < 0.000 | *p* = 0.008 | *p* = 0.053 | | | | |
| Pockets SRP | 85.8 ± 14.2 | 4.83 ± 1.18 | 3.41 ± 1.63 | | | | |
| | 38.3 ± 23.9 | 4.0 ± 0.99 | 2.6 ± 1.34 | | | | |
| | *p* < 0.000 | *p* = 0.005 | *p* = 0.037 | | | | |

Lack of significant differences in the final bleeding on probing (BoP) values (*p* = 0.73), average PD (0.53), and average clinical attachment loss (CAL) (0.63) between the pockets treated with SRP+A and SRP alone.

The results of microbiological observations are presented in Table 2. In general, no statistically significant differences in the assessment of the related variables (the same pockets before and after treatment), and the nonrelated variables (symmetric pockets before and after treatment using both methods) were found. The only exceptions concerned the pockets additionally treated with antibiotic gel. Two months post-treatment, a significant increase of *Treponema denticola* and decrease of *Micromonas micros* as well as a significantly higher amount of *Capnocytophaga gingivalis* in relation to the control pockets were found. For the latter, reduction in the amount of *Fusobacterium nucletaum* and *Micromonas micros* was considerably similar to the significant decrease after SRP.

**Table 2.** Results of microbiological tests.

| Type of Bacteria | SRP+A Treatment | SRP Treatment | P for Rows |
|---|---|---|---|
| TBC before after | 455,603,800 ± 774,832,582 225,700,000 ± 160,305,299 T26, *p* = 0.88 | 138,810,000 ± 184,812,292 400,090,000 ± 710,668,522 T16, *p* = 0.24 | U = 50 ns U = 50 ns |
| P.g. before after | 55,050 ± 105,157 94,000 ± 239,040 T8, *p* = 0.6 | 54,328 ± 146,867 182,514 ± 365,513 T5, *p* = 0.25 | U = 43 ns U = 47 ns |
| T.d. before after | 4802 ± 5793 36,188 ± 79,754 T3, *p* = 0.036 | 6456 ± 11,483 35,460 ± 50,421 T9, *p* = 0.11 | U = 42 ns U = 48.5 ns |
| T.f. before after | 11,372 ± 23,587 7996 ± 10,964 T11, *p* = 0.33 | 40,227 ± 106,011 16,657 ± 29,545 T21, *p* = 0.86 | U = 49.5 ns U = 47 ns |
| A.a before after | Liczba wykryć-3 Liczba wykryć-2 | Liczba wykryć-2 Liczba wykryć-1 | |
| F.n. before after | 6800 ± 14,499 466 ± 615 T10, *p* = 0.26 | 8257 ± 18,822 716 ± 1320 T6, *p* = 0.051 | U = 35 ns U = 43 ns |
| P.i. before after | 43,404 ± 106,163 199,610 ± 497,956 T8, *p* = 0.6 | 61,153 ± 144,493 23,765 ± 69,086 T5, *p* = 0.25 | U = 46 ns U = 40 ns |
| P.m. before after | 2877 ± 2644 631 ± 767 T3, *p* = 0.036 | 5914 ± 11,514 589 ± 616 T6, *p* = 0.051 | U = 49.5 ns U = 48.5 ns |
| E.n. before after | Liczba wykryć-1 Liczba wykryć-1 | Liczba wykryć-1 Liczba wykryć-2 | |
| C.g. before after | 14,636 ± 37,211 5439 ± 4962 T25, *p* = 0.8 | 5878 ± 6165 2050 ± 1796 T8, *p* = 0.086 | U = 41.5 ns U = 22.5 |

The influence of the current periodontitis classification on the assessed amounts of bacteria is presented in Table 3. A small number of analyzed samples and lack of detection of particular pathogens in the pockets influenced considerable values of standard deviations and the significance of the intergroup differences. Quantitative differentiation of the studied compounds of the pocket microbiome was influenced to a larger extent by the degrees of periodontitis before the treatment (considerably higher number of TBC in degree B, as well as higher—though marginally—numbers of *Porphyromonas gingivalis*, *Treponema denticola*, *Prewvotella intermedia*, and *Capnocytophaga* in relation to C).

**Table 3.** Results of microbiological tests depending on the stadium and the degree of periodontitis.

| Type of Bacteria | Stage II and III vs. IV | p | Grade B vs. C | p |
|---|---|---|---|---|
| TBC before after | 404,295,571 ± 660,136,448 vs. 473,333,333 ± 493,301,180 386,071,429 ± 595,735,933 vs. 142,150,000 ± 114,175,369 | 0.23 0.16 | **410,078,571 ± 656,851,542 vs. 33,839,667 ± 42,535,311** 369,428,571 ± 602,988,291 vs. 180,983,333 ± 102,863,996 | **0.048** 0.74 |
| P.g. before after | 64,141 ± 145,840 vs. 32,633 ± 50,957 73,439 ± 230,011 vs. 289,500 ± 417,614 | 0.59 0.59 | 68,641 ± 144,276 vs. 22,133 ± 52,856 79,653 ± 229,047 vs. 275,000 ± 427,446 | 0.3 0.71 |
| T.d. before after | 5731 ± 9219 vs. 5392 ± 8920 40,956 ± 74,652 vs. 23,850 ± 35,928 | 0.9 0.48 | 7438 ± 10,099 vs. 1408 ± 2223 41,263 ± 74,510 vs. 23,133 ± 36,272 | 0.3 0.54 |
| T.f. before after | 32,595 ± 90,773 vs. 9943 ± 14,466 11,238 ± 21,684 vs. 14,867 ± 25,083 | 0.62 0.77 | 34,431 ± 90,504 vs. 5660 ± 8455 11,224 ± 21,730 vs.15,133 ± 24,856 | 0.71 0.97 |
| A.a before after | Liczba wykryć-3 vs. 2 Liczba wykryć-2 vs. 1 | | Liczba wykryć-3 vs. 2 Liczba wykryć-2 vs. 1 | |
| F.n. before after | 9221 ± 19,245 vs. 3578 ± 5346 565 ± 856 vs. 652 ± 1405 | 0.59 0.65 | 9264 ± 19,221 vs. 3478 ± 5442 457 ± 873 vs. 903 ± 1318 | 0.97 0.26 |
| P.i. before after | 40,398 ± 92,291 vs. 80,000 ± 186,333 148,125 ± 424,800 vs. 26,667 ± 65,320 | 0.53 0.2 | 66,669 ± 14,447 vs. 18,700 ± 44,736 158,075 ± 423,223 vs. 3450 ± 7209 | 0.34 0.33 |
| P.m. before after | 2948 ± 2751 vs. 7773 ± 14,874 695 ± 676 vs. 412 ± 702 | 0.77 0.2 | 5547 ± 9698 vs. 1708 ± 1891 544 ± 564 vs. 765 ± 937 | 0.36 0.77 |
| E.n. before after | Liczba wykryć-2 vs. 0 Liczba wykryć-3 vs. 0 | | Liczba wykryć-2 vs. 0 Liczba wykryć-3 vs. 0 | |
| C.g. before after | 12,261 ± 31,319 vs. 5580 ± 7049 3264 ± 2426 vs. 4867 ± 6646 | 0.45 0.87 | 13,679 ± 31,124 vs. 2272 ± 3107 3336 ± 2366 vs. 4698 ± 6746 | 0.09 0.65 |

## 5. Discussion

Currently, only one study assessing clinical and microbiological effectiveness of locally applied piperacillin with tazobactam in patients with periodontitis is available.

Lack of placebo application may constitute a limitation of the study; both the person who collected the material and the laboratory staff were subjected to a blind test.

In our own studies on clinical parameters, such as bleeding on probing (BoP) index, periodontal pocket depth (PPD), and location of the connective tissue attachment (CAL), the following results were obtained after 2 months: BoP was reduced (−49.2% vs. 47.5% for SRP), the average PPD was shallowed (0.82 vs. 0.83 mm for SRP), and additionally, attachment location in the control pockets was improved (−0.81 mm). On the other hand, after 2 months of treatment, no differences in the assessed clinical parameters between the two groups were observed. In the studies conducted by Lauenstein et al. [9,10], in which piperacillin with tazobactam were applied locally, similar reduction in the values of clinical parameters was obtained. The difference in PPD measurements between the initial examination and the examination conducted in the control group after 26 weeks was 1.8 mm (SE ± 0.3; 95% CI 1.2, 2.3; $p < 0.001$). The average difference in PPD between the initial value and the 26th week in the study group was 1.5 mm (SE ± 0.2; 95% CI 1.1, 2.0; $p < 0.001$), and the statistical analysis did not show any differences in the study groups in terms of PPD values both at the beginning of the study and in its 26th week. Moreover, in the case of the parameter assessing inflammation (BoP) no statistically significant differences between the groups were proven, neither at the beginning of the study nor during the final

examination after 26 weeks. The fact that the study included smokers may constitute a disruptive factor [10]. The obtained reduction in the clinical parameters confirms the necessity of conducting nonsurgical treatment, especially in terms of reducing periodontal pocket depth and the location of connective tissue attachment, and confirms the gold-standard status of the scaling and root planning procedure in periodontal treatment [11–13]. Mechanical cleaning is necessary in order to improve clinical parameters, and the applied preparations can only support and consolidate the treatment results [14,15].

With regard to the microbiological tests, there are currently no reports confirming effective reduction of periopathogens in periodontal pockets after local application of piperacillin in combination with tazobactam in patients with moderate and advanced periodontitis. No statistically significant differences between the studied groups were found in our own studies. After local application of Gelcide in combination with SRP, a significant increase in *Treponema denticola* and decrease in *Micromonas micros* as well as a considerably higher number of *Capnocytophaga gingivalis* were found in the microbiological test in relation to the control pockets. After application of SRP alone, a decrease in the number of *Fusobacterium nucletaum* and *Micromonas micros* was observed. In the studies conducted by Lauenstein et al. [10], reduction in pathogens (including *Porphyromonas gingivalis, Tanerella forsythia,* or *Agregatibacter actinomycetemcomitans*) after local application of piperacillin with tazobactam, in comparison with application of only mechanical cleaning, was obtained in both groups after 26 weeks. On the other hand, significant reduction of *Fusobacterium nucleatum, P. micra*, and *T. denticola* was obtained in the group in which the treatment was combined with local antibiotic therapy.

Due to bacteria organization within the biofilm present in the periodontal pockets and the presence of periopathogenic bacteria in the tissues of the host, eradication of periopathogens and long-term improvement of clinical and in particular, microbiological parameters, is virtually impossible. In the case of local application of doxycycline, the results of clinical and microbiological tests are inconclusive. Reports on considerable improvement of clinical parameters and reduction in periopathogens after SPR combined with intrapocket administration of antibiotic [16,17] are available. Long-term observations, however, do not indicate considerable benefits [18]. At the same time, some scientists do not prove significant differences in results with regard to the independent use of SRP in periodontal disease treatment [19]. Difficulties in obtaining complete reduction of periopathogens, regardless of the type of therapy, were also presented in the studies by Mobelli et al. [20] after local application of tetracycline fibers [21]. The study included seventeen patients, with whom microbiological analysis of subgingival samples collected mesially–distally from 852 areas at the beginning and after a month of using the preparation was conducted. In the basic study, 46 samples, from 10 positive individuals, showed positive results for *P. gingivalis*; 82 samples, from 5 individuals, were also identified as positive in terms of *A. actinomycetemcomitans*. The presence of *A. actinomycetemcomitans* and *P. gingivalis* was not confirmed in the material collected from 3 patients. Microbiological tests conducted one month after periodontal treatment showed that 89% of areas which initially showed positive results in terms of pathogens were negative, while 16 areas which were initially identified as negative showed positive results. With regard to *A. actinomycetemcomitans*, 77% of areas which were identified as positive in terms of bacteria presence in the first test were later identified as negative, but 5 areas which were initially identified as negative showed positive results. During the examinations of patients whose results of the microbiological test remained positive, another attempt at applying local treatment using tetracycline fibers was made. Despite another attempt at treatment, microbiological tests still showed the presence of *P. gingivalis* in 5 individuals; and in 4 patients, the results for *A. actinomycetemcomitans* remained positive. Those 9 patients were finally subjected to systemic antibiotic treatment ($3 \times 250$ mg metronidazole and $3 \times 375$ mg amoxicillin/per day for 7 days). Despite all the efforts, after 3 months *P. gingivalis* was again detected in 3 individuals and *A. actinomycetemcomitans* was isolated from 1 area.

Organization of periopathogens within the biofilm, their diversity, and ability to quickly recolonize bacteria still constitutes a great therapeutic challenge. Nonsurgical methods (SRP) are still the basis for treatment of periodontium diseases, and local or general application of antibiotic therapy is

still ineffective in terms of periopathogen elimination [22]. In summary, this study showed similar improvement of clinical parameters in patients treated with SRP combined with single administration of local antibiotic (piperacillin/tazobactam) or without it, without showing statistically significant differences in the microbiological tests results. Piperacilin in combination with tazobactam is used primarily for the general treatment of nosocomial infections. Clinical data indicate that the combination of piperacillin/tazobactam is effective in the treatment of moderate to severe polymicrobial infections, including intra-abdominal, skin, and soft-tissue and lower respiratory tract infections. It shows high efficiency in the case of infections caused by *Escherichia coli,* many *Bacteroides* and *Klebsiella species, Staphylococcus aureus,* and *Haemophilus influenzae*, with low resistance after the treatment. The combination of piperacillin and tazobactam in dentistry has been not well documented. There are no medical studies that investigate the effectiveness against pathogens that cause periodontitis. Despite the broad spectrum of gram-negative bacteria, there is no evidence of the effects on periopathogens. It should be emphasized that, in the case of topical application in periodontal pockets, there is currently no available literature. It is required to conduct clinical trials confirming the spectrum of the drug's action on periopathogens, together with correlation of the clinical results. In vitro studies should also be considered to systematize and confirm the efficacy of piperacillin/tazobactam on specific periopathogens.

Further clinical studies on larger groups of patients and long-term observations of the obtained results or changes occurring in them in order to optimize the algorithms of nonsurgical treatment of periodontal diseases and minimize application of general antibiotic therapy is required.

**Author Contributions:** Author A.S.-J. caried SRP, author J.Z. carried drug administration and article review and editing, author M.S. carried study design and draft preparation, author T.K. carried examination and project supervision.

**Funding:** This research received no external funding

**Conflicts of Interest:** The authors declare no conflict of interest.

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
