# Peer review of "New Local Drug Delivery with Antibiotic in the Nonsurgical Treatment of Periodontitis—Pilot Study"

_applsci, doi:10.3390/app9235077_

Round 1

Reviewer 1 Report

This manuscript presents data to evaluate whether the combination of the SRP with an antibiotic administered to periodontal pockets is a promising alternative protocol of non-surgical periodontal treatment. The strength of this study is that patients were selected with strictly flow under proper approval by professional committee. The groups were designed well. Although there is no significant clinical and microbiological differences were found after additional administration of gel with antibiotics, this study provides as a reference for further clinical studies to optimize the periodontal disease therapy. There is no major concern. But several minor concerns are needed to be addressed: 1. the potential mechanisms are supposed to be well discussed in the Discussion. 2. the accuracy of the text needed to be check.

Author Response

the potential mechanisms are supposed to be well discussed in the Discussion

The text has been reworded.

Piperacilin in combination with tazobactam is used primarily for the general treatment of nosocomial infections. Clinical data indicate that the combination of piperacillin / tazobactam is effective in the treatment of moderate to severe polymicrobial infections, including intra-abdominal, skin and soft-tissue and lower respiratory tract infections. It shows high efficiency in the case of infections caused by Escherichia coli, many Bacteroides and Klebsiella species, Staphylococcus aureus, and Haemophilus influenzae with low resistance after the treatment. The combination of piperacillin and tazobactam in dentistry has been not well  documented. There are no medical studies,  investigete the effectiveness  against pathogens that cause periodontitis. Despite of the broad spectrum of on gram-negative bacteria, there is no evidence of the effects on  periopathogens. It should be emphasized that in the case of topical application in periodontal pockets, there is currently no available literature, despite one. It is required to conduct clinical trials confirming the spectrum of the drug's action on periopathogens, together with correlation with the clinical results. In vitro studies should also be considered to systematize and confirm the efficacy of piperacillin / tazobactam on specific periopathogens.

The accuracy of the text needed to be check.

The text has been reworded.

Reviewer 2 Report

The novelty of this study is great, Gelcide is a new and interesting material, with relevant references.

The lot of patients is limited (10) and  and I think it is irrelevant even for a pilot study. If "The protective layer which, according to the manufacturer, was supposed to protect the patients from rapid decrease in drug concentration in the pocket within 8 - 10 days was unnoticeable", why the evaluation was made 2 month later?

In row 12: Material and methods: 10 patients aged 24-56 years (mean 39.5) with chronic periodontitis. In row 86: 10 patients aged 24 to 26 (the average age was 39.5, 5 males, 5 females).Seems another lot of patients.

Author Response

The lot of patients is limited

            A relatively small research group results from the need to use many excluded factors that significantly limited the time of research. We are aware that appropriate research requires a higher number of patients to improve treatment outcomes.

"The protective layer which, according to the manufacturer, was supposed to protect the patients from rapid decrease in drug concentration in the pocket within 8 - 10 days was unnoticeable", why the evaluation was made 2 month later?

The presence of the protective layer was not the target of the study, it was assessed only at check-up visits, 8 weeka after application,  and these parameters were not included in the results. The purpose of the study was the results of the microbiological effectiveness after 2 mounth.

The period of 2 months for the microbioological examination was chosen due to the possible recolonization of periopathogens after this time and to evaluate the microbiological long-term effects.

In row 12: Material and methods: 10 patients aged 24-56 years (mean 39.5) with chronic periodontitis. In row 86: 10 patients aged 24 to 26 (the average age was 39.5, 5 males, 5 females).

It is an editorial error.

10 patients aged 24 to 56 (the average age was 39.5, 5 males, 5 females) with general periodontitis (loss of CAL on interproximal surfaces, CAL>30% of teeth, with moderately symmetric bone loss in pantomographic image) were qualified for the study.

Reviewer 3 Report

Manuscript title:

New Local Drug Delivery with Antibiotic in the Non3 Surgical Treatment of Periodontitis- Pilot Study

The authors summarized that clinical and microbiological evaluation of periodontal pockets 2 months after single application of a gel containing piperacillin and tazobactam in relation to the non-surgical treatment procedure (SRP).

The authors concluded that no significant clinical and microbiological differences were found after additional 26 administration of gel with piperacillin and tazobactam in relation to SRP in the preliminary study.

There some major concerns about the style and methods. The reviewer thinks the manuscript can be published after the major revision of below things.

Please revised as follows; Line 86: 10 patients aged 24 to 26 56 Line 203: can Please provide the information of operator. Ex) periodontist or not periodontist. The difference of personal treatment skill has a great effect for the result. The examinations determined parameters; Did the authors investigated the tooth mobility? All patients have no systemic disease? Please provide the more information. This paper is negative paper. Please provide the more information why no significant difference? Ex) such as molecular mechanism or Bacterial flora changes. Please comply with the author guidelines. Ex) Font, reference style

Author Response

Line 86: 10 patients aged 24 to 26 56 Line

It is an editorial error.

10 patients aged 24 to 56 (the average age was 39.5, 5 males, 5 females) with general periodontitis (loss of CAL on interproximal surfaces, CAL>30% of teeth, with moderately symmetric bone loss in pantomographic image) were qualified for the study. 

203: can

The text has been reworded.

Please provide the information of operator.

All of operators are periodontists and the text has been reworded.

Did the authors investigated the tooth mobility?

We did not investigated the tooth mobility

All patients have no systemic disease?

All patients have no systemic disease, were non-smokers. The exluded criteria were: diabetes mielitus, hypertention, smoking, using antibiotics within the period of 6 months prior to the study, no allergy to penicillin in medical history.

 Please provide the more information. This paper is negative paper. Please provide the more information why no significant difference? 

No positive result is also a result. Piperacilin in combination with tazobactam is used primarily for the general treatment of nosocomial infections. Clinical data indicate that the combination of piperacillin / tazobactam is effective in the treatment of moderate to severe polymicrobial infections, including intra-abdominal, skin and soft-tissue and lower respiratory tract infections. It shows high efficiency in the case of infections caused by Escherichia coli, many Bacteroides and Klebsiella species, Staphylococcus aureus, and Haemophilus influenzae with low resistance after the treatment. The combination of piperacillin and tazobactam in dentistry has been not well  documented. There are no medical studies,  investigete the effectiveness  against pathogens that cause periodontitis. Despite of the broad spectrum of on gram-negative bacteria, there is no evidence of the effects on  periopathogens. It should be emphasized that in the case of topical application in periodontal pockets, there is currently no available literature, despite one. It is required to conduct clinical trials confirming the spectrum of the drug's action on periopathogens, together with correlation with the clinical results. In vitro studies should also be considered to systematize and confirm the efficacy of piperacillin / tazobactam on specific periopathogens.

 Please comply with the author guidelines. Ex) Font, reference style

The text has been reworded.

Round 2

Reviewer 2 Report

I have no more suggestions for the authors